# Measurement Properties of the Dutch Multifactor Fatigue Scale in Early and Late Rehabilitation of Acquired Brain Injury in Denmark

**DOI:** 10.3390/jcm12072587

**Published:** 2023-03-29

**Authors:** Frederik Lehman Dornonville de la Cour, Trine Schow, Tonny Elmose Andersen, Annemarie Hilkjær Petersen, Gry Zornhagen, Annemarie C. Visser-Keizer, Anne Norup

**Affiliations:** 1Cervello, 2800 Kongens Lyngby, Denmark; 2Department of Psychology, University of Southern Denmark, 5230 Odense, Denmark; 3Center for Rehabilitation of Brain Injury, 2300 Copenhagen, Denmark; 4Center for Communication Disorders, The Capital Region of Denmark, 2750 Ballerup, Denmark; 5Center for Rehabilitation, University Medical Center Groningen, 9700 RB Groningen, The Netherlands; 6Neurorehabilitation Research and Knowledge Centre, Rigshospitalet, 2600 Glostrup, Denmark

**Keywords:** fatigue, brain injuries, stroke, psychometrics, neurological rehabilitation, patient reported outcome measures

## Abstract

Fatigue is a major issue in neurorehabilitation without a gold standard for assessment. The purpose of this study was to evaluate measurement properties of the five subscales of the self-report questionnaire the Dutch Multifactor Fatigue Scale (DMFS) among Danish adults with acquired brain injury. A multicenter study was conducted (*N* = 149, 92.6% with stroke), including a stroke unit and three community-based rehabilitation centers. Unidimensionality and measurement invariance across rehabilitation settings were tested using confirmatory factor analysis. External validity with Depression Anxiety Stress Scales (DASS-21) and the EQ-5D-5L was investigated using correlational analysis. Results were mixed. Unidimensionality and partial invariance were supported for the Impact of Fatigue, Mental Fatigue, and Signs and Direct Consequences of Fatigue, range: RMSEA = 0.07–0.08, CFI = 0.94–0.99, ω = 0.78–0.90. Coping with Fatigue provided poor model fit, RMSEA = 0.15, CFI = 0.81, ω = 0.46, and Physical Fatigue exhibited local dependence. Correlations among the DMFS, DASS-21, and EQ-5D-5L were in expected directions but in larger magnitudes compared to previous research. In conclusion, three subscales of the DMFS are recommended for assessing fatigue in early and late rehabilitation, and these may facilitate the targeting of interventions across transitions in neurorehabilitation. Subscales were strongly interrelated, and the factor solution needs evaluation.

## 1. Introduction

Fatigue is a widespread issue following acquired brain injury (ABI), and about half of stroke survivors experience post-stroke fatigue [1,2]. The course of fatigue post-injury remains unclear [3,4,5,6], but fatigue can be persistent and long lasting with adverse effects on functional outcome [7,8]. Fatigue interferes with quality of life, participation in everyday activities, and return to work [9,10,11].

The experience of fatigue is inherently subjective, and no consensus exists regarding a definition of fatigue [12]. Fatigue may be defined as “difficulty in initiation of or sustaining voluntary activities” [13] (p. 978), or more broadly as “the awareness of a decreased capacity for physical and/or mental activity due to an imbalance in the availability, utilization, and/or restoration of resources needed to perform activity” [14] (p. 46). The distinction between mental and physical fatigue is common, although conceptually debated [15], as fatigue may be related to specific domains of activities or task performance [12]. For instance, fatigue may respond to mental exertion or manifest as difficulties with engagement in cognitively demanding tasks, e.g., working memory or sustained attention. The pathophysiological mechanisms are elusive [16,17], and the genesis and chronicity of fatigue following ABI have multiple interacting causes and contributors at biological, psychological, and social levels [18,19]. Consequently, planning of treatment and rehabilitation requires a detailed assessment of the nature of fatigue, including characteristics, precursors, consequences, and management strategies.

A wide range of self-report scales of fatigue are used for ABI populations [20], but only few were developed specifically for these patient groups [21,22]. Although generic scales are useful for comparisons across populations [23], disease-specific characteristics of ABI may be confused with the effects of fatigue when responding to a generic questionnaire [21,22,24,25]. For example, items on the Multidimensional Fatigue Inventory [26] address fatigue in terms of impact on functioning and everyday activities, which can also be affected by paresis, attention deficits, or other common sequelae of ABI. Consequently, characteristics of the patient group may affect the validity of the scale, which requires consideration in scale development and application.

The Dutch Multifactor Fatigue Scale (DMFS) was designed to assess the multidimensional nature of fatigue following ABI specifically [27]. Items were derived based on patient interviews and evaluated among individuals in the chronic stage of ABI, i.e., at least six months post-injury [27]. DMFS offers a detailed account of fatigue following ABI, and a promising feature is the subscale Coping with Fatigue, which addresses management strategies used by patients to cope with the limitations posed by their symptoms of fatigue. This aspect is not well addressed by existing scales in stroke [28]. Implementing coping strategies is a central goal of rehabilitation, and it may be beneficial to assess recovery and intervention effects in terms of the ability to cope with challenges posed by fatigue instead of—or in addition to—changes in the intensity or impact of fatigue. Further, the multidimensional account of fatigue offered by DMFS may assist the clinical examination of fatigue to facilitate targeted treatment and rehabilitation programs.

DMFS demonstrated good internal consistency, convergent validity with another measure of fatigue, and divergent validity with measures of mood and self-esteem [27]. To the authors’ knowledge, no psychometric evaluation of DMFS has been conducted since the initial validation. Further, test validity has not been evaluated for individuals earlier than six months post-injury. Early planning of treatment is important for recovery outcomes, and the consistent use of assessment instruments across transitions in rehabilitation promotes interpretability in the evaluation of fatigue during recovery. Better clinical assessment can potentially help the patient understand and cope with fatigue in everyday life.

The aim of the present study was to evaluate measurement properties of the Danish version of DMFS among adults with ABI. More specifically, objectives were to test (i) unidimensionality of subscales, (ii) measurement invariance across sub-acute vs. community-based rehabilitation settings, and (iii) external validity with symptoms of depression, anxiety, and stress and quality of life. The study was part of a larger validation project on DMFS, and parallel research elucidated response processes to DMFS using cognitive interviewing [25].

## 2. Materials and Methods

A prospective cross-sectional multicenter validation study was conducted. The study was conducted in accordance with the Declaration of Helsinki [29]. An ethical request was submitted to The Danish National Committee on Health Research Ethics, but ethical approval was not required according to Danish national legislation, as the study was identified as a health scientific survey study not involving human biological data.

### 2.1. Recruitment Procedures

Participants were recruited consecutively from September 2018 to February 2020 in four neurorehabilitation units in Denmark, including (i) three rehabilitation centers in a community setting and (ii) one sub-acute stroke rehabilitation unit in a hospital department. Inclusion criteria were: (1) ≥18 years old, (2) ABI, (3) Danish speaking, and (4) able to provide informed consent. Individuals were excluded in the case of (1) progressive brain disease, (2) mild traumatic brain injury (based on medical records), or (3) overt cognitive difficulties interfering with participation. The sample was intended to represent clinical practice, and thus no criterion was defined regarding time since injury or emotional disorders such as depression.

At community-based rehabilitation centers, therapists, neuropsychologists, and master’s students in psychology screened individuals in rehabilitation for eligibility. The members of a psychosocial intervention group at one of the rehabilitation centers were also invited to participate. At the stroke unit, a physiotherapist and an occupational therapist screened patients at admission to the unit. All participants provided written informed consent. The target sample size was 150 based on recommendations for factor analysis [30].

### 2.2. Outcome Measures

The DMFS [27] consists of 38 items distributed on five subscales: Impact of Fatigue (11 items), Signs and Direct Consequences of Fatigue (9), Mental Fatigue (7), Physical Fatigue (6), and Coping with Fatigue (5). Responses were offered on a 5-point Likert-type response scale, anchored with the terms *no, I strongly disagree*; *I mostly disagree*; *neutral*; *I mostly agree*; and *yes, I strongly agree*, scored 1 to 5. Nine items were reverse coded prior to analysis. Greater scores indicate more problems, i.e., more severe fatigue or less ability to cope with limitations posed by fatigue.

The short version of the Depression Anxiety Stress Scales (DASS-21) was used to assess negative emotional states [31,32]. DASS-21 comprises three subscales with seven items each. Greater scores indicate more emotional distress. Cronbach’s α was 0.84, 0.79, and 0.87 for the Depression, Anxiety, and Stress scales, respectively.

The EQ-5D-5L was used to assess health-related quality of life [33]. The EQ-5D-5L contains five dimensions (mobility, self-care, usual activities, pain/discomfort, anxiety/depression), each of which is rated on five levels from *no problems* to *extreme problems*. Ratings represent a unique health state, which was mapped to an index value with higher values representing better health-related quality of life [34].

### 2.3. Data Collection

Data were collected by clinicians in routine practice using paper-based materials. At the stroke unit, data were collected as close as possible to discharge. If needed, items were read aloud to inpatients, and responses were recorded with the aid of the therapist and a separate visualization of the response options. The DASS-21 and EQ-5D-5L were not administered to inpatients to limit response burden. The DMFS took about 15 ± 5 min to complete.

### 2.4. Data Analysis

DMFS subscales were analyzed separately. Descriptive statistics on item and composite scores were conducted, including inter-item correlations (using polychoric correlations, *r*_pc_) and corrected item-total correlations (using polyserial correlations, *r*_ps_) [35]. Monotonicity was examined by visual inspection of scatterplots of item scores vs. rest-scores. Confirmatory factor analysis (CFA) was conducted on raw data using WLSMV, which does not assume normality and is robust with small samples [36]. Global fit was evaluated using the adjusted χ^2^ test statistic, the root mean square error of approximation (RMSEA), the Comparative Fit Index (CFI), and the Tucker–Lewis Index (TLI). Adequacy of model fit was based on the following criteria: RMSEA ≤ 0.10 and CFI and TLI ≥ 0.90, with RMSEA in the range of 0.08–0.10 indicating mediocre fit [36]. Reliability was estimated using the ω total coefficient [37] for categorical items [38]. For comparison with other studies, Cronbach’s α was also reported.

Measurement invariance was tested across rehabilitation settings (sub-acute vs. community-based) using multi-group CFA with ordinal indicators. For each subscale, a series of three hierarchically nested models with increasingly restrictive equality constraints (i.e., configural, thresholds, and loadings) was evaluated following guidelines by Svetina et al. [39] based on the approach of Wu and Estabrook [40]. Relative model fit between nested models was evaluated using the strictly positive Satorra–Bentler scaled difference χ^2^ test statistic [41] and ∆RMSEA and ∆CFI. Criteria for determining non-invariance based on fit indices have not been evaluated adequately for models with ordinal indicators [39]. Thus, although based on continuous indicators, the following criteria recommended for small and unequal sample sizes were used: ∆CFI ≤ −0.005 and ∆RMSEA ≥ 0.010 indicate non-invariance [42].

In the case of misfit or non-invariance of CFA models, parameter estimates, modification indices, and residuals were examined to guide any post-hoc respecifications or testing of partial invariance (i.e., releasing constraints contributing to misfit). Models were deemed partially invariant if ≤20% of the constrained parameters were freed [43]. Further details on CFA procedures are provided in Appendix B.

External validity was examined using Pearson’s product-moment correlation coefficient. The following hypotheses were tested:

**H1.** 
*Large positive correlations (r~0.65) among Impact of Fatigue, Mental Fatigue, and Signs and Direct Consequences of Fatigue, and moderate-to-large positive correlations (40 < r < 50) with Physical Fatigue;*


**H2.** 
*Small positive correlations (r~0.15) between Coping with Fatigue vs. other subscales;*


**H3.** 
*Moderate positive correlations (r~0.40) between DMFS (excluding Coping with Fatigue) vs. DASS-21;*


**H4.** 
*Moderate negative correlations (r~−0.30) between DMFS (excluding Coping with Fatigue) vs. EQ-5D-5L health index.*


Hypotheses 1–3 were based on findings in previous research [27]. Hypothesis 4 was based on findings indicating associations between fatigue and health-related quality of life [11,44,45].

Analyses were conducted in R version 4.2.0 [46] using the *psych* [47] package for descriptive and correlational analyses, the *lavaan* [48] and *semTools* [49] packages for CFA, and the *MBESS* [50] package for reliability analyses. Missing data were handled using listwise deletion due to low missing data rates (<5% for all items). The study was reported using the COSMIN Reporting Guideline [51].

## 3. Results

Out of 160 recruited, 8 participants were excluded (cerebral tumor, *n* = 6; transient ischemic attack, *n* = 2). In addition, three participants did not complete the assessment. Thus, the final sample size was 149 (*n* = 100 in community-based rehabilitation, *n* = 49 in sub-acute rehabilitation). Table 1 presents sample characteristics. Notably, the majority had a stroke (92%), and time since injury varied substantially across rehabilitation settings (Figure 1). In community-based rehabilitation, median time since injury was 9 months, ranging from 1–161 months. For stroke inpatients, median time since injury was 13 days, ranging from 5–47 days.

### 3.1. Unidimensionality

Statistics on DMFS subscales are reported in Table 2. Item statistics are provided in the Appendix A, including inter-item correlations (Appendix A), corrected item-total correlations and missing responses (Appendix A), item response distributions (Appendix A), monotonicity plots (Appendix A), and standardized factor loadings (Appendix A). Multiple items exhibited skewed response distributions, predominantly negatively skewed (Appendix A).

#### 3.1.1. Impact of Fatigue

Inter-item correlations were positive. Item 24, “I don’t need to have a rest to make it through the day”, was weakly associated with most other items, *r*_pc_ range from 0.10–39, and the total score, *r*_ps_ = 27. A one-factor model fit the data well, and reliability was good to excellent (Table 2). Factor loadings were significant and large (λ range: 0.65–0.86), except for item 24 (λ_24_ = 0.33). No violations of monotonicity were detected.

#### 3.1.2. Signs and Direct Consequences of Fatigue

Inter-item correlations were positive, and the one-factor model exhibited adequate fit (Table 2). Factor loadings ranged from 0.45–0.72, all significant, and reliability was adequate to good (Table 2). No serious violations of monotonicity were detected.

#### 3.1.3. Mental Fatigue

Inter-item correlations were positive. The one-factor model provided good fit to data, and reliability was good (Table 2). All factor loadings were salient and significant, ranging from 0.47–0.86. No violations of monotonicity were evident.

#### 3.1.4. Physical Fatigue

Inter-item correlations were positive, but the factor model exhibited misfit (Table 2). Factor loadings were salient and significant. The largest modification index was for the error covariance of items 5 and 9 (δ_5,9_ = 18.7). Both items had relatively large loadings, λ = 0.79 and 0.80, compared to the other indicators (λ range: 0.41–0.63). Items 5, “I feel physically fit”, and 9, “I am in good physical condition”, are conceptually related, which justified testing local dependence in a nested model. Freeing δ_5,9_ improved model fit significantly, scaled ∆χ^2^(1) = 19.77, *p* < 0.001. Global fit of the adjusted model was good, χ^2^(8) = 13.29, *p* = 0.10, RMSEA = 0.07, 90% CI [0.00, 0.10], CFI = 0.99, TLI = 0.97, and reliability was poor, ω = 0.68. The covariance of items 5 and 9 was salient and significant, δ_5,9_ = 0.57, *p* < 0.001, and factor loadings attenuated (λ = 0.54 and 0.56). Overall, the results indicated local dependence among items 5 and 9, violating the assumptions of unidimensionality, and further analyses on Physical Fatigue were terminated, including invariance testing and external validity.

#### 3.1.5. Coping with Fatigue

Two item pairs were negatively correlated. The one-factor model on Coping with Fatigue provided inadequate fit, and reliability was poor (Table 2). Consequently, further analyses were terminated.

### 3.2. Measurement Invariance across Rehabilitation Settings

Table 3 shows fit and comparison statistics of the multi-group CFA models tested for invariance. Standardized factor loadings of the final models are provided in Appendix A.

#### 3.2.1. Impact of Fatigue

The baseline model converged on an improper solution with a non-positive definite model matrix (minimum determinant = −4.48). Item 24, “I don’t need to have a rest to make it through the day”, was unrelated to the latent variable among inpatients, λ_24_ = −0.01, *p* = 0.96, whereas all other loadings were salient and significant. Thus, item 24 was omitted in a revised model. Fit indices of the 10-item baseline model supported configural invariance. Further, both threshold and metric invariance were supported, and the final model provided adequate fit (Table 3). Factor loadings were positive, salient, and significant.

#### 3.2.2. Signs and Direct Consequences of Fatigue

Fit of the baseline model was mediocre. Threshold invariance was supported, but equality constraints on factor loadings caused misfit based on ∆CFI (Table 3). Modification indices identified item 7, “Emotional issues make me tired”, as the largest source of misfit. Releasing this constraint (1/8 (12.5%) constrained loadings freed) improved model fit, and partial metric invariance was supported (Table 3). The factor loading was larger in community-based settings, λ_7_ = 0.70, *p* < 0.001, compared to sub-acute settings, λ_7_ = 0.30, *p* = 0.004, indicating that item 7 performs worse among inpatients. Remaining factor loadings of the partial invariant model were positive, salient, and significant.

#### 3.2.3. Mental Fatigue

Configural invariance and threshold invariance were supported (Table 3). However, equality constraints on loadings caused a substantial decrease in CFI (−0.006), and modification indices identified item 3, “I can follow conversations without getting tired”, as the largest cause of misfit. Partial metric invariance was supported (Table 3) with this constraint freed (1/6 (16.7%) constrained loadings freed). Item 3 performed poorly in the sub-acute setting, λ_3_ = 0.19, *p* = 0.12, while performing well in community-based settings, λ_3_ = 0.58, *p* < 0.001. Remaining factor loadings were positive, salient, and significant.

### 3.3. External Validity

In community-based rehabilitation settings, DMFS subscales were positively intercorrelated in expected relative magnitudes (Table 4). However, correlations among Impact of Fatigue, Signs and Direct Consequences of Fatigue, and Mental Fatigue (*r* range: 0.77–0.81) were stronger than expected in Hypothesis 1, indicating that these subscales address very closely related aspects. Hypothesis 2, regarding Coping with Fatigue, was not tested due to violations of unidimensionality.

Impact of Fatigue, Signs and Direct Consequences of Fatigue, and Mental Fatigue were all moderately to strongly related with DASS-21 (*r* range: 0.28–0.62). Correlations with the Depression and Anxiety scales ranged from *r* = 0.32–0.38 and *r* = 0.28–0.32, respectively, as expected in Hypothesis 3. Correlations with the Stress scale were relatively stronger, *r* = 0.47–0.62.

Finally, Impact of Fatigue, Signs and Direct Consequences of Fatigue, and Mental Fatigue exhibited moderate negative correlations with EQ-5D-5L health index; *r* ranged from −0.40 to −0.45, indicating associations with health-related quality of life, as expected in Hypothesis 4.

## 4. Discussion

This multicenter study provides mixed results on the validity of DMFS among adults with ABI, the majority with stroke. Three subscales performed adequately in both sub-acute and community-based rehabilitation settings, namely, Impact of Fatigue, Signs and Direct Consequences of Fatigue, and Mental Fatigue, and Physical Fatigue and Coping with Fatigue exhibited psychometric issues. Findings add to the scarce evidence on measurement properties of self-report fatigue scales in ABI populations and supplement previous research on DMFS.

DMFS addresses challenges related to the assessment of fatigue following ABI [27]. DMFS was developed specifically for this patient group, and items address a broad range of characteristics named by patients, including the nature and impact of fatigue, co-occurring symptoms, and ways of coping and managing fatigue. On this basis, DMFS is a promising instrument for the clinical assessment of fatigue in brain injury rehabilitation and may assist targeting of treatment to individual needs. Based on present findings, three subscales of DMFS are recommended, namely, Impact of Fatigue, Signs and Direct Consequences of Fatigue, and Mental Fatigue. However, the three subscales were strongly interrelated, and larger studies are needed to evaluate the optimal factorial structure of DMFS. The subscales were associated with emotional symptoms, consistently with previous research on DMFS [27] and other measures of fatigue [52,53,54,55]. Interestingly, DMFS correlated more strongly with symptoms of stress than depression and anxiety, and the substantial overlap of these affective symptoms with fatigue needs consideration in clinical assessment and research designs.

Despite being developed for the chronic stage of ABI (i.e., ≥6 months post-injury), present findings indicate that DMFS may be used in sub-acute stages as well. A few items performed worse among inpatients, however, including items 3, 7, and 24. Although fatigue is frequently reported during hospitalization [2,3,4], these patients are yet to experience the degree to which fatigue interferes with daily activities after discharge and how fatigue responds to the demands associated with pre-injury activity levels and personal identity, e.g., work and family roles, daily chores, etc. Inpatients may lack knowledge about fatigue and have difficulties making an accurate judgement about their post-injury level of functioning. On this basis, the patient’s narrative of fatigue may evolve during the recovery process, affecting how patients respond to self-report outcome measures. For example, patients in early stages of recovery may report that fatigue is unpredictable and occurs without obvious explanation, and in later stages, some may learn to identify subtle signs of fatigue and respond to these in time to avoid overexertion. In addition, complicated and cognitively demanding questions may be more challenging to process for hospitalized patients, and inpatients may need more support when responding to DMFS (e.g., reading questions aloud). These factors may explain why some items performed worse among inpatients. Nevertheless, evidence supports the use of Impact of Fatigue (excluding Item 24), Signs and Direct Consequences of Fatigue, and Mental Fatigue across rehabilitation settings, and these subscales may assist the clinical assessment across transitions in rehabilitation and facilitate appropriate management and treatment of fatigue.

On the other hand, Physical Fatigue and Coping with Fatigue demonstrated issues. Items 5 and 9 on Physical Fatigue exhibited local dependence, indicating that these items were influenced by an extraneous variable in addition to the latent variable. Consequently, unidimensionality was not supported, and reliability was poor. This finding is consistent with parallel research revealing that respondents tend to refer to general health problems, physical fitness, and physical sequelae rather than fatigue when responding to these items [25]. These patterns in response processes may explain the local dependence evident in this study.

Coping with Fatigue demonstrated poor fit and reliability, and some items were unrelated to each other. Thus, items seem to reflect different aspects, which compromises interpretation of the sum score. The concept of coping is complex and multifaceted, including cognitive, behavioral, and social aspects, and coping involves multiple strategies and approaches such as problem-focused, emotion-focused, and meaning-focused coping [56]. Generic coping inventories such as COPE [57] comprises multiple dimensions, addressing distinct coping strategies. Likewise, various strategies for managing fatigue may be used independently of each other, and, as argued by Billings and Moos [58], the efficient use of one coping strategy may reduce the need to use others, which may limit the internal consistency of a coping scale. In addition, reasons for maladaptive coping vary, e.g., lack of knowledge about fatigue and how to respond to fatigue, insufficient awareness of early signs of fatigue, poor planning, and unhelpful cognitive schemes such as thinking that one must finish tasks at hand before taking a rest. Finally, some strategies may be helpful in some circumstances and not others. From clinical experience, patients are often flexible rather than definitive in their use of strategies. Now and then, patients may deliberately prioritize attending a personally valued activity above the cost of overexertion and associated symptoms. In contrast, some items address behavior in definitive terms, e.g., item 16, “I avoid becoming overtired”, which may confuse responders and mislead interpretation.

Assessing coping with fatigue remains a challenge. Nevertheless, there is a need for instruments addressing these aspects, as coping and management strategies are essential to rehabilitation and treatment of fatigue following ABI. Future research may benefit from identifying additional thoughts and behaviors that patients use to manage fatigue in daily life, e.g., to reduce distress associated with fatigue, promote wellbeing, and prevent overexertion, and to operationalize these for an inventory of coping with fatigue following ABI. Future efforts also need to consider the flexibility and conflicting interests of patients in managing fatigue in daily life. In addition to the retrospective account of questionnaires, momentary assessment and interview techniques may also be useful, and addressing activity levels (e.g., vocational and leisure activities) as well as symptoms may be integral to evaluate recovery and effectiveness of treatment and rehabilitation targeting fatigue.

### Limitations

The present findings relate to the Danish version of DMFS, and it is uncertain to what extent these findings reflect translation variations pertinent to the Danish version or general features in common with the English and the original (Dutch) version. Subtle deviations in wording may alter the meaning and interpretation of items and affect measurement properties [25]. Considering the ambiguity of fatigue, variations across languages need to be considered carefully. In addition, any cultural differences in the experience and self-report of fatigue need consideration. Cultural aspects of fatigue are poorly understood, although previous research reported lower levels of fatigue in Asian populations compared to Western ones [1]. Consequently, validation of the English version is recommended to compare with results on the Danish version.

Characteristics of the sample differ somewhat from the initial validation of DMFS by Visser-Keizer et al. [27], e.g., regarding etiology and time since injury. These aspects may affect the performance of DMFS and account for differences in results between the studies. Eligibility criteria were defined with the aim of representing patients encountered in clinical practice for a more direct translation of study results into the clinic. However, characteristics of participants need consideration in the generalization of results, e.g., regarding age. Notably, the majority of the sample had stroke, and caution is needed when generalizing results to other types of ABI.

Finally, the sample was relatively small considering the statistical analyses employed. The precision of parameter estimates and the performance of fit indices are sensitive to sample size [59], and both need to be interpreted cautiously. However, all response categories were observed (Appendix A), and collapsing categories with few observations in post-hoc analyses did not affect results substantially (see Appendix A). Further, the estimation method (WLSMV) is robust with small samples [36], and parameters were estimated without problems of non-convergence, improper solutions, or Heywood cases. Nevertheless, a replication of results is recommended, and a larger sample will enable more complex models to evaluate the proposed factorial structure, which is warranted considering the strong correlations among subscales. Additional properties are also to be examined in the next steps of validation such as test–retest reliability and responsiveness to change. Any modifications to the DMFS based on the results of this study and parallel research [25], including a potential short scale, also need to be tested in future research.

## 5. Conclusions

Few self-report measures of fatigue have been developed and validated in ABI populations, and this study provides evidence on measurement properties of DMFS among adults with ABI in early and late rehabilitation settings. Three out of five subscales are recommended for assessing fatigue in sub-acute and community-based rehabilitation settings and may be used across transitions in rehabilitation. In contrast to most other scales available, DMFS offers a detailed account of multiple aspects of fatigue following ABI. Thus, DMFS may be useful for characterizing problems of fatigue and targeting treatment to individual needs. However, present results indicate that the original subscales are closely interrelated and question whether the current factor solution is optimal.

## Figures and Tables

**Figure 1 jcm-12-02587-f001:**
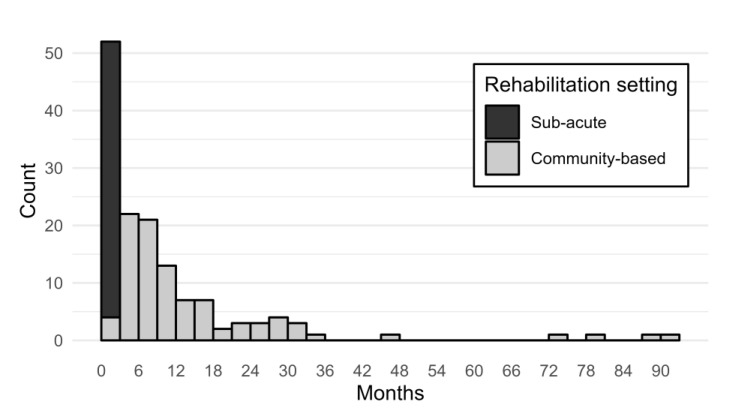
Time since injury at assessment. Note. The histogram shows frequency distributions of months since injury among participants in sub-acute (*n* = 49) and community-based rehabilitation (*n* = 100), respectively. Bins span 3 months. Two outliers at 153 and 161 months, respectively, are not shown.

**Table 1 jcm-12-02587-t001:** Characteristics of participants.

Characteristic	Community-Based (*n* = 100)	Sub-Acute (*n* = 49)	Full Sample (*N* = 149)
Sex, *n* (%)			
Female	37 (37.0)	19 (38.8)	56 (37.6)
Male	63 (63.0)	30 (61.2)	93 (62.4)
Age, *M* (*SD*)	54.3 (10.7)	66.7 (12.2)	58.4 (12.6)
Education, *M* (*SD*) ^1^	15.0 (3.3) ^2^	14.4 (4.0)	14.8 (3.6) ^3^
Days since injury, *M* (*SD*)	540.8 (801.3) ^4^	17.2 (10.5) ^5^	367.4 (669.4) ^6^
Type of injury, *n* (%)			
Stroke	89 (89.0)	49 (100.0)	138 (92.6)
Traumatic brain injury	5 (5.0)	0 (0.0)	5 (3.4)
Other ^7^	6 (6.0)	0 (0.0)	6 (4.0)
Type of stroke, *n* (%)			
Ischemic	59 (66.3)	33 (67.3)	92 (66.7)
Hemorrhagic	26 (29.2)	12 (24.5)	38 (27.5)
Both	3 (3.4)	0 (0.0)	3 (2.2)
Missing data	1 (1.2)	4 (8.2)	5 (3.6)
Previous brain injury, *n* (%)	16 (16.0)	9 (18.4)	25 (16.8)
Missing data	4 (4.0)	7 (14.3)	11 (7.4)

Note. ^1^ Reported in years. ^2^ *n* = 99. ^3^ *n* = 148. ^4^ *n* = 97. ^5^ *n* = 48. ^6^ *n* = 145. ^7^ Includes central nervous system infection, aneurism surgery, anoxia, and hydrocephalus.

**Table 2 jcm-12-02587-t002:** Statistics on subscales of Dutch Multifactor Fatigue Scale.

Scale	Descriptive Statistics			Global Model Fit					Reliability	
	*N*	*M* (*SD*)	Range	χ^2^ (*df*)	*p*	CFI	TLI	RMSEA [90% CI]	ω [95% CI]	α
IF	147	38.7 (10.4)	11–55	73.37 (44)	0.004	0.981	0.977	0.068 [0.039, 0.094]	0.90 [0.86, 0.92]	0.88
SC	149	28.9 (7.6)	9–45	51.01 (27)	0.003	0.944	0.925	0.078 [0.044, 0.110]	0.80 [0.71, 0.85]	0.77
MF	148	25.2 (6.3)	7–35	23.11 (14)	0.058	0.985	0.978	0.067 [0.000, 0.113]	0.83 [0.76, 0.87]	0.81
PF	148	17.4 (5.4)	6–30	40.64 (9)	<0.001	0.918	**0.863**	**0.155** [0.108, 0.204]	0.76 [0.65, 0.82]	0.71
CF	148	15.3 (3.9)	7–25	22.06 (5)	0.001	**0.806**	**0.613**	**0.152** [0.091, 0.220]	0.46 [0.15, 0.61]	0.46

Note. Bold indicates misfit. Models were fitted to raw data using WLSMV in *R* v. 4.2.0 using *lavaan*. CFI = comparative fit index; TLI = Tucker–Lewis index; RMSEA = root mean square error of approximation; CI = confidence interval; ω = McDonald’s omega total coefficient for categorical items; α = Cronbach’s coefficient alpha; IF = Impact of Fatigue; SC = Signs and Direct Consequences of Fatigue; MF = Mental Fatigue; PF = Physical Fatigue; CF = Coping with Fatigue.

**Table 3 jcm-12-02587-t003:** Measurement invariance of Dutch Multifactor Fatigue Scale across rehabilitation settings.

Scale	Equality Constraints	Invariance	Global Model Fit					Model Comparisons				Freed
			χ^2^ (*df*)	*p*	CFI	TLI	RMSEA	∆χ^2^ (∆*df*)	*p*	∆CFI	∆RMSEA	
IF	Form	-	-	-	-	-	-					
IF-10	Form	Full	91.74 (70)	0.042	0.988	0.984	0.065					
	τ	Full	116.57 (90)	0.031	0.985	0.985	0.064	24.76 (20)	0.211	−0.003	−0.001	
	τ + λ	Full	116.56 (99)	0.110	0.990	0.991	0.049	5.82 (9)	0.758	0.005	−0.015	
SC	Form	Full	81.53 (54)	0.009	0.939	0.919	0.083					
	τ	Full	101.37 (72)	0.013	0.935	0.935	0.074	16.81 (18)	0.537	−0.004	−0.009	
	τ + λ	Rejected	114.50 (80)	0.007	0.924	0.931	0.077	12.04 (8)	0.150	**−0.011**	0.003	
		Partial	106.75 (79)	0.021	0.939	0.944	0.069	7.52 (7)	0.377	0.004	−0.005	λ_7_
MF	Form	Full	47.37 (28)	0.013	0.972	0.957	0.097					
	τ	Full	61.08 (42)	0.029	0.972	0.972	0.079	10.64 (14)	0.714	0.000	−0.018	
	τ + λ	Rejected	71.52 (48)	0.015	0.966	0.970	0.082	9.07 (6)	0.170	**−0.006**	0.003	
		Partial	61.08 (47)	0.081	0.979	0.982	0.064	3.29 (5)	0.655	0.005	−0.015	λ_3_

Note. Bold indicates invariance. Models were fitted to raw data using multi-group confirmatory factor analysis on community-based (*n* = 100) vs. sub-acute (*n* = 49) rehabilitation settings. Estimation: WLSMV in *R* v. 4.2.0 using *lavaan*. CFI = comparative fit index; TLI = Tucker–Lewis index; RMSEA = root mean square error of approximation; IF = Impact of Fatigue (11 indicators); IF-10 = Impact of Fatigue w/o item 24 (10 indicators); SC = Signs and Direct Consequences of Fatigue (9 indicators); MF = Mental Fatigue (7 indicators); τ = threshold; λ = factor loading.

**Table 4 jcm-12-02587-t004:** Descriptive statistics and pairwise correlations in community-based rehabilitation (*n* = 100).

Variable (Range)	*n*	*M*	*SD*	1	2	3	4	5	6	7	8	9
DMFS												
1. IF (11–55)	99	38.72	10.79	–								
2. SC (9–45)	100	29.62	7.53	0.79 ***	–							
3. MF (7–35)	100	25.52	6.19	0.81 ***	0.77 ***	–						
4. PF (6–30)	99	16.39	5.15	0.64 ***	0.57 ***	0.49 ***	–					
5. CF (5–25)	99	15.22	4.14	0.26	0.30 *	0.22	0.24	–				
DASS-21												
6. Depression (0–42)	97	6.85	7.93	0.36 **	0.38 **	0.32 *	0.37 **	0.09	–			
7. Anxiety (0–42)	97	4.91	7.20	0.29 *	0.32 *	0.28	0.35 **	0.09	0.55 ***	–		
8. Stress (0–42)	98	11.20	9.91	0.47 ***	0.62 ***	0.52 ***	0.42 ***	0.32 *	0.68 ***	0.62 ***	–	
EQ-5D-5L												
9. Index (−0.624–1.000)	98	0.77	0.14	−0.45 ***	−0.41 ***	−0.40 **	−0.45 ***	−0.06	−0.29 *	−0.19	−0.31 *	–

Note. Correlations were computed using Pearson’s product-moment correlation coefficient, and *p*-values were adjusted using Holm correction. IF = Impact of Fatigue; SC = Signs and Direct Consequences of Fatigue; MF = Mental Fatigue; PF = Physical Fatigue; CF = Coping with Fatigue. * *p* < 0.05. ** *p* < 0.01. *** *p* < 0.001.

## Data Availability

The data presented in this study are available as summary data in the online Appendix A. Raw data are not available due to privacy.

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
