# Peer review of "Measurement Properties of the Dutch Multifactor Fatigue Scale in Early and Late Rehabilitation of Acquired Brain Injury in Denmark"

_jcm, 2023, doi:10.3390/jcm12072587_

Round 1
Reviewer 1 Report
Statistical analysis to validate is essential for verifying the reliability of this study. The authors described the analysis method in the manuscript and added detailed supplementary material. However, there is no file in the link (www.mdpi.com/xxx/s1) to view this supplement, so I cannot judge the appropriateness of this paper. Please double-check the supplementary materials and links.
Author Response
Thank you very much for your comments.
I am sorry the supplementary material was not available. I uploaded the file with the manuscript in the submission process, so I guess they should be available for review somehow.
I expect that the link in the manuscript will be updated during production, if the manuscript is accepted for publication. But I will check with the editor.
Reviewer 2 Report
I read tha Manuscript titled '' Measurement Properties of the Dutch Multifactor Fatigue Scale in Early and Late Rehabilitation of Acquired Brain Injury in Denmark'' carefully.
I find the Manuscript is a well written one with a high novelty of an overage importance. I wonder to which institution were the ethical request was submitted. I recommend to mention the institution's name in the methods section. I dont have any further comments.
Author Response
Thank you very much for your valuable comments.
We have elaborated on the details regarding ethical approval, including the name of the institution we have been in contact with. Please find the changes on page 2 in the revised manuscript.
Reviewer 3 Report
This is an interesting study of validation of the Dutch Multifactor Fatigue Scale in subacute and chronic rehabilitation of acquired brain injury in Denmark that included mainly patients with ischemic stroke, the manuscript is well-written and adequately analyzed, the additional limitation is the age of the patients, the type of cooperative patients and the lack of control of potential enhancing drugs to improve the fatigue in the evaluation. Otherwise if adequate.
Author Response
Thank you for your comments.
Regarding your point on the additional limitations, we have elaborated on the study limitations regarding generalization of results in the Limitations section, Page 10, Lines 391-395:
"Eligibility criteria were defined with the aim of representing patients encountered in clinical practice for a more direct translation of study results into the clinic. However, characteristics of participants need consideration in the generalization of results, e.g., regarding age. Notably, the majority of the sample had stroke, and caution is needed when generalizing results to other types of ABI."
Reviewer 4 Report
The purpose with this study was to evaluate measurement properties of the five subscales of the Dutch Multifactor Fatigue Scale (DMFS), with 5 subscales, among Danish adults with acquired brain injury. A multicentre study with mainly stroke patients were included, early and late after a stroke.
Unidimensionality and measurement invariance was tested confirmatory factor analysis. Also external validity in relation to depression, anxiety and quality of life.
Three of the subscales of DMFS are recommended for assessing fatigue in early and late rehabilitation.
The aim was to “evaluate measurement properties of the Danish 82 version of DMFS among adults with ABI”.
Comments:
The intention is good, finding scales for fatigue after acquired brain injuries. This is an important and common problem and needs to be acknowledged, even in the early phase after an ABI, as well as in follow up. It is a merit having uses this scale in the early phase.
The manuscript is well written. I have no specific knowledge in the statistics used for the evaluation of the scale, but have some general questions of the scale and the study.
Why not an ethical approval? It is reported “ethics approval was not required by the local ethics committee due to the study design”. A more comprehensive motivation to this is needed. In addition, is this in conflict with this journal?
Three of the subscales are recommended and suggested to “be useful for characterizing problems of fatigue and targeting treatment to individual needs. Is there any cut-off scores indicating significant fatigue compared to normality? How compared to healthy controls, what is normal range for the subscales?
Why not also compare with other fatigue scales for the validity evaluation?
Can DMFS be used after other acquired brain injuries or neurological illnesses? The intention was to include acquired brain injuries, but most was stroke patients, is DMFS also useful for other brain injuries?
Is it important to have a specific scale for stroke or do other patients groups suffering from fatigue report similar problems?
The cooping subscale – should it be included?
Any problems with the invers questions for fatigue people? Your personal experience? Commonly they have problems with flexibility while suffering from fatigue, and invers questions can be interpreted wrong or not be understood. Is there a problem with the construction of the scale?
Figure 1. Time since injury at assessment. Add more information. I guess it is the frequency of participants this figure shows?
Author Response
Thank you for your thorough review and valuable comments to the manuscript. Please find our response in the attachment.
